# Seasonal Monitoring of Volatiles and Antioxidant Activity of Brown Alga *Cladostephus spongiosus*

**DOI:** 10.3390/md21070415

**Published:** 2023-07-21

**Authors:** Sanja Radman, Martina Čagalj, Vida Šimat, Igor Jerković

**Affiliations:** 1Department of Organic Chemistry, Faculty of Chemistry and Technology, University of Split, R. Boškovića 35, 21000 Split, Croatia; 2Department of Marine Studies, University of Split, R. Boškovića 37, 21000 Split, Croatia; mcagalj@unist.hr (M.Č.); vida@unist.hr (V.Š.)

**Keywords:** brown algae, GC-MS, HS-SPME, hydrodistillate, aliphatic compounds, terpenes, UHPLC-ESI-HRMS, fatty acid amides, xanthophylls, antioxidant activity, UAE, MAE

## Abstract

*Cladostephus spongiosus* was harvested once a month during its growing season (from May to August) from the Adriatic Sea. Algal volatile organic compounds (VOCs) were obtained by headspace solid-phase microextraction (HS-SPME) and hydrodistillation (HD) and analysed by gas chromatography and mass spectrometry (GC-MS). The effects of air drying and growing season on VOCs were determined. Two different extraction methods (ultrasound-assisted extraction (UAE) and microwave-assisted extraction (MAE)) were used to obtain ethanolic extracts of *C. spongiosus*. In addition, the seasonal antioxidant potential of the extracts was determined, and non-volatile compounds were identified from the most potent antioxidant extract. Aliphatic compounds (e.g., pentadecane) were predominantly found by HS-SPME/GC-MS. Hydrocarbons were more than twice as abundant in the dry samples (except in May). Aliphatic alcohols (e.g., hexan-1-ol, octan-1-ol, and oct-1-en-3-ol) were present in high percentages and were more abundant in the fresh samples. Hexanal, heptanal, nonanal, and tridecanal were also found. Aliphatic ketones (octan-3-one, 6-methylhept-5-en-2-one, and (*E*,*Z*)-octa-3,5-dien-2-one) were more abundant in the fresh samples. Benzene derivatives (e.g., benzyl alcohol and benzaldehyde) were dominant in the fresh samples from May and August. (*E*)-Verbenol and *p*-cymen-8-ol were the most abundant in dry samples in May. HD revealed aliphatic compounds (e.g., heptadecane, pentadecanal, (*E*)-heptadec-8-ene, (*Z*)-heptadec-3-ene), sesquiterpenes (germacrene D, epi-bicyclosesquiphellandrene, gleenol), diterpenes (phytol, pachydictyol A, (*E*)-geranyl geraniol, cembra-4,7,11,15-tetraen-3-ol), and others. Among them, terpenes were the most abundant (except for July). Seasonal variations in the antioxidant activity of the ethanolic extracts were evaluated via different assays. MAE extracts showed higher peroxyl radical inhibition activity from 55.1 to 74.2 µM TE (Trolox equivalents). The highest reducing activity (293.8 µM TE) was observed for the May sample. Therefore, the May MAE extract was analysed via high-performance liquid chromatography with high-resolution mass spectrometry and electrospray ionisation (UHPLC-ESI-HRMS). In total, 17 fatty acid derivatives, 9 pigments and derivatives, and 2 steroid derivatives were found. The highest content of pheophorbide *a* and fucoxanthin, as well as the presence of other pigment derivatives, could be related to the observed antioxidant activity.

## 1. Introduction

The genus *Cladostephus* currently consists of five taxonomically recognised species. These species are cosmopolitan brown algae that grow on rocks in the intertidal zone and at depths of up to six metres, mainly in temperate seas [1]. *Cladostephus spongiosus* occurs in the Adriatic Sea [2]. Its thalli are usually between 3 and 20 cm long, with the longest thallus present in summer, during its growing season, when the sea temperature is the highest. In winter, when the sea temperature is at its lowest, only small specimens 3 cm in length are observed [3].

Several studies have confirmed the bioactivity of compounds obtained from this species. The extracts from *C. spongiosus* have been reported to exhibit antioxidant, antibacterial, anticandidal, cytoprotective, insecticidal, and amoebicidal activities [4,5,6,7,8]. It has been previously reported that the bioactivity of the extracts could be affected by the season of algal harvest and extraction method [9,10,11]. However, to our knowledge, no studies on these effects have been performed for *C. spongiosus*. Air-dried samples of *C. spongiosus* harvested from the Algerian coast were extracted and the extracts were analysed by HPLC and GC-MS. Fucoxanthin, apo-9′-fucoxanthinone, apo-13′-fucoxanthinone, and loliolid were identified in the samples [12]. The major compounds identified in the acetone extracts of *C. spongiosus* were 4-hydroxy-4-methylpentan-2-one, *n*-hexadecanoic acid, and propenylguaiacol, while octadeca-9,12,15-trienoic acid ethyl ester, hexadecanoic acid ethyl ester, and 3,7,11,15-tetramethylhexadec-2-en-1-ol were the major compounds in the ethanol extracts [8].

Whitfield et al. [13] isolated volatile compounds from *C. spongiosus* harvested in Australia using combined steam distillation solvent extraction with pentane/diethyl ether. Overall, 2,6-dibromophenol, 2-bromophenol, 2,4,6-tribromophenol, 4-bromophenol, and 2,4-dibromophenol were identified in the samples using gas chromatography and mass spectrometry (GC-MS). In addition, the lipid phase of freeze-dried samples of *C. spongiosus* collected off the Algarve coast in Portugal was extracted and analysed by GC-MS. The authors reported 31.74% of saturated fatty acids, 12.15% of monounsaturated fatty acids, and 56.11% of polyunsaturated fatty acids [14]. Rubiño et al. [15] determined the profile of volatile compounds in *C. spongiosus* harvested in Spain using headspace solid-phase microextraction (HS-SPME). The authors tentatively identified pent-1-en-3-one, pent-2-en-1-ol, hexanal, hexan-1-ol, tribromomethane, 1-iodo-3-methylbutane, and oct-1-en-3-one. To our knowledge, there are no studies on the changes of volatile organic compounds (VOCs) during the growth period of *C. spongiosus*. In the present study, VOCs were separated and analysed from air-dried and fresh *C. spongiosus* samples from the Adriatic Sea using HS-SPME and hydrodistillation (HD). These techniques were chosen because they allow the isolation headspace, volatile and semi-volatile compounds. Algal samples were collected during the growing season from May to August. The study objectives were: (1) to isolate VOCs from *C. spongiosus*, (2) to determine the chemical composition of obtained VOCs, (3) to determine the effects of air drying and growing season on VOCs composition, (4) to determine the seasonal antioxidant potential of *C. spongiosus* ethanolic extracts using different assays, (5) to compare the ultrasound-assisted extraction (UAE) and microwave-assisted extraction (MAE) to produce the extracts with higher antioxidant activity, and (6) to identify the non-volatile compounds in the most potent ethanolic extract.

## 2. Results and Discussion

VOCs were obtained using two methods (HS-SPME and HD) followed by GC-MS analysis. A wide range of VOCs were identified (headspace, as well as low to medium volatile compounds). Antioxidant assays were performed to analyse the seasonal potential of the extracts. High-performance liquid chromatography with high-resolution mass spectrometry and electrospray ionisation (UHPLC-ESI-HRMS) analysis was performed on the most potent antioxidant extract to identify present non-volatile compounds.

### 2.1. Headspace Volatilome Variations of C. spongiosus

Two fibres of different polarity, divinylbenzene/carboxene/polydimethylsiloxane (DVB/CAR/PDMS, f1) with medium polarity and polydimethylsiloxane/divinylbenzene (PDMS/DVB, f2) with high polarity, were used to study the headspace composition.

Aliphatic compounds accounted for the majority of compounds identified in all samples from four consecutive months. Their lowest percentage was in the air-dried sample in May (36.90%, f1; 35.93%, f2) and the highest in the air-dried sample in August (91.14%, f1; 89.30%, f2). When extracted with f1 (fibre of medium polarity), hydrocarbons were dominant in both the fresh and dry samples from June (23.02%; 47.01%) and from August (26.28%; 65.94%) and in the dry sample from July (47.01%) (Figure 1). Extraction with the more polar f2 fibre resulted in a dominance of hydrocarbons only in the dry samples from June (35.70%), July (35.96%), and August (65.16%) (Figure 2). In the dry samples, hydrocarbons were more than twice as high as in the fresh samples, except in May, when they were more abundant in the fresh samples. Pentadecane dominated with the highest abundance in August samples (43.88%, f1; 35.68%, f2) (Table 1 and Table 2). This result is consistent with previous findings on the predominance of pentadecane in brown algae [16]. Other abundant hydrocarbons were pentadec-1-ene and heptadecane. Hydrocarbons are likely formed as oxidative degradation products of lipids [17].

Aliphatic alcohols were present with high percentages in all months studied, especially when extracted with the fibre f2 (Figure 2), and they were more abundant in the fresh samples. The short-chain alcohols ethanol, hexan-1-ol, octan-1-ol, and oct-1-en-3-ol were the most abundant (Table 1 and Table 2). Previous studies showed that the oxylipin oct-1-en-3-ol is a defence compound in marine algae [18,19,20].

Hexanal, heptanal, nonanal, and tridecanal constituted the majority of aldehydes (Table 1 and Table 2). The algal aldehydes are formed through degradation of fatty acids, which can occur either via oxidation or via the enzymatic action of lipoxygenases [21]. Hexanal and heptanal are mainly derived from linoleic acid [22,23]. Nonanal could originate from ω9 monounsaturated fatty acids (MUFAs) and ω6 polyunsaturated fatty acids (PUFAs), such as linoleic acid [24]. The fatty aldehyde tridecanal could be derived from long-chain fatty acids. Aldehydes obtained with the polar fibre f2 were more abundant in the air-dried samples in all months.

Aliphatic ketones, such as octan-3-one, 6-methylhept-5-en-2-one, and (*E*,*Z*)-octa-3,5-dien-2-one, were more abundant in fresh samples (f1, f2). The fresh sample from June contained the highest percentage of aliphatic ketones (16.31%, f1; 20.58%, f2) (Table 1 and Table 2), which play the role of pheromones in marine algae [12].

Benzene derivatives group dominated the fresh samples from May (33.86%, f1; 21.42%, f2), July (16.72%, f1; 27.87%, f2), and August (21.35%, f1; 35.34%, f2), as can be seen in Figure 1 and Figure 2. Benzyl alcohol and benzaldehyde formed the majority of benzene derivatives. Benzaldehyde was the predominant compound in all fresh samples obtained by f2 (Table 2) and in all fresh samples (except June) obtained by f1 (Table 1). In contrast, benzyl alcohol was more abundant in the dry samples. The volatile benzene derivatives can be formed from phenyalanine when the side chain of a carbon skeleton shortens by C2-unit. This reaction most commonly occurs via the β-oxidative pathway [25]. The loss of benzaldehyde during air-drying could be due to its high volatility [26,27,28].

Terpenes, monoterpenes, and sesquiterpenes showed the largest proportion (Figure 1 and Figure 2) in May dry sample (15.32%, f1; 13.94%, f2), mainly with the high abundance of monoterpene alcohols (*E*)-verbenol and *p*-cymen-8-ol (Table 1 and Table 2).

In the group of other compounds carboxylic acids, dictyopterenes, and C_13_-norisoprenoides were abundant. Carboxylic acids such as hexanoic and heptanoic acids contributed strongly to the increase in other compounds abundance, especially in the dry May sample.

### 2.2. Statistical Analysis of the Headspace VOCs

The results of the principal component analyses (PCA) are shown in Figure 3a–d and Figure 4. PCA was used to describe the variations among the dominant volatiles (>2%) in relation to the sampling month, material preparation (fresh or dry), and fibre.

The correlation plot and score plot of the dominant components from data obtained by f1 and f2 fibres are shown in Figure 3a–d, respectively. The first two PCs of the data obtained by f1 fibre described 53.4% of the initial data variability. The highest variable contribution to PC1 was observed for tridecanal, hexanal, pentadec-1-ene, and ß-ionone, while ethanol, hexan-1-ol, octan-1-ol, and phenol contributed to PC2.

The first two PCs in the data obtained by f2 fibre described 62.2% of the initial data variability. The abundance of two aldehydes (nonanal and tridecanal) contributed to PC1, while alcohols (ethanol and hexan-1-ol) and phenol had the highest contribution to the PC2. The distribution of the samples in the multivariate space showed no significant difference between the fibres. However, a clear separation between the fresh and dried samples was obtained (Figure 3b,d). For both f1 and f2, the dry samples were positioned on the left part of the multivariate space and the fresh samples, taken in May, were separated in the bottom right part of the score plot. Interestingly, the sampling month showed no effect on the dry samples, while in fresh samples there are similarities between June, July, and August, while May was clearly separated in the upper right.

Figure 4a,b shows the PCA of the data for f1 and f2 analysed together. The initial data variability of the first two PCs described 52%; however, the fresh and dried samples were clearly separated based on the higher content of (*E*,*Z*)-octa-3,5-dien-2-one and benzyl alcohol, as well as low content of pentadecane in the fresh samples. The samples from both fibres were segregated according to the sampling month, with clear vertical distribution from May to August.

### 2.3. Hydrodistillation Obtained Volatilome Variations of C. spongiosus

In the hydrodistillate, the percentages of the identified compounds ranged from 81.21% (Fr_July) to 89.11% (Dr_August) of the total compounds detected. These compounds were classified into four different groups: aliphatic compounds, sesquiterpenes, diterpenes, and others (Figure 5). Terpenes, especially sesquiterpenes and diterpenes, were the most abundant in all samples (36.09%, Fr_August—52.08%, Fr_May), except in the fresh sample in July (33.13%). Sesquiterpenes were more abundant only in May’s fresh sample (30.47%), mainly because of the high proportion of germacrene D (9.63%) and epi-bicyclosesquiphellandrene (4.49%) and gleenol (11.58%) (Table 3). All three detected sesquiterpenes were the predominant sesquiterpenes in the hydrodistillate of brown alga *T. atomaria* which belongs to the same subclass Dictyotophycidae [29]. When observing the seasonal changes in *H. scoparia*, which belongs to the same order of Sphacelariales, the highest proportion of germacrene D and gleenol in the hydrodistillate of the fresh sample was also found in May and later decreased each month until September [30]. Diterpenes were more abundant in the remaining samples (Figure 5). Diterpene alcohols phytol (4.42%, Fr_July—18.73%, Dr_August), pachydictyol A (3.18%, Fr_August—12.44%, Dr_May), (*E*)-geranyl geraniol (1.40%, Fr_May—6.77%, Dr_July), and cembra-4,7,11,15-tetraen-3-ol (4.92%, Fr_May—16.04%, Dr_July) dominated. The lowest percentage of detected diterpenes was always found in the fresh samples and the highest in dry samples. Phytol content was the highest in August. A similar behaviour was observed in *H. scoparia* where phytol was the predominant compound [30]. Phytol and its derivatives have a wide range of bioactive functions such as antioxidant, cytotoxic, anti-inflammatory, antimicrobial, autophagy- and apoptosis-inducing, immune- and metabolism-modulating, and anxiety-relieving activities [31]. Cembra-4,7,11,15-tetraen-3-ol was also found to be a very dominant terpene in *D. dichotoma* and *H. scoparia*, as was pachydictyol A in *D. dichotoma* [30,32]. Cembranoid-type diterpenes have good antitumor, antimicrobial, and neuroprotective activity [33], and pachydictyol A can be used as an antifouling paint, among other applications [34]. Monoterpenes were present only in the dry sample in May (< 1%) and in both the fresh and dry samples in June (<0.1%). In general, terpenes are bioactive compounds and, as such, play an important role in human health, as well as food preservatives [35].

Fatty acid ethyl ester (FAEE) ethyl icosanoate, assigned to the group of others, was the dominant compound in all air-dried samples (12.85%, Dr_May—30.74%, Dr_July) and in the fresh samples in June (30.84%) and July (34.51%). Naturally occurring FAEEs as well as fatty acid methyl esters (FAMEs) could potentially be used as biofuels [36].

The saturated hydrocarbon heptadecane and aldehyde pentadecanal, as well as the unsaturated hydrocarbons (*E*)-heptadec-8-ene and (*Z*)-heptadec-3-ene, were the major aliphatic compounds with the highest percentage in August, contributing to the representation of the aliphatic compounds (Figure 5). All were most abundant in the fresh sample in August except heptadecane which was not detected but was most abundant in the dry sample in the same month (Table 3).

### 2.4. Statistical Analysis of the VOCs Obtained via Hydrodistillation

The PCA results for VOCs of fresh and air-dried *C. spongiosus* obtained via hydrodistillation are shown in Figure 6a,b. The first two PCs described 76.3% of the initial data variability. The correlation loadings of the first two PCs (Figure 6a) showed correlations between germacrene D and gleenol, and between aliphatic compounds ((*E*)-heptadec-8-ene, (*Z*)-heptadec-3-ene and pentadecanal). Epi-bicyclosesquiphellandrene and cembra-4,7,11,15-tetraen-3-ol were the variables with the highest variable contributions, based on the correlations. The PC2 was associated with germacrene D, phytol, and pachydictyol A abundance in the samples. The score plot (Figure 6b) showed the position samples in the multivariate space of the first two PCs. There was a clear separation between June and July samples in the centre right part of the plot, and segregation between May and August samples.

June and July samples, characterized with higher content of cembra-4,7,11,15-tetraen-3-ol and ethyl icosanoate, and lower percentage of germacrene D and gleenol were positioned in the right part of the plot. The vertical distribution between May and August samples was related to differences in germacrene D, gleenol, pachydictyol A, and phytol abundance. There were more differences in seasonal variation between fresh samples. The distribution was along the PC1 axis and cannot be related to one group of compounds but was in the relation to the abundance of epi-bicyclosesquiphellandrene, cembra-4,7,11,15-tetraen-3-ol, ethyl icosanoate and pentadecanal, while the distribution along the PC2 axis was related to terpenes (germacrene D, phytol and pachydictyol A) abundance in the samples. No correlation was found between the compounds’ content and temperature change.

### 2.5. Antioxidant Activity of Ethanol Extracts In Vitro

Seasonal variations in antioxidant activity of *C. spongiosus* extracts produced by MAE and UAE were evaluated by ORAC, FRAP, and DPPH assays (Figure 7). All extracts from MAE showed higher activity. The ORAC results ranged from 54.9 ± 1.5 to 69.8 ± 1.9 µM TE for UAE extracts and from 55.1 ± 1.1 to 74.2 ± 1.0 µM TE for MAE extracts. The highest peroxyl radical inhibition activity was observed for the samples harvested in May. The FRAP results ranged from 157.4 ± 3.9 to 238.5 ± 21.4 µM TE for UAE extracts, and from 176.4 ± 4.7 to 293.8 ± 9.2 µM TE for MAE extracts. The highest reducing activity was also observed for the samples harvested in May. The scavenging ability of DPPH radical ranged from 18.2 ± 0.7 to 27.3 ± 0.7% for UAE extracts, and from 24.4 ± 1.8 to 32.1 ± 0.5% for MAE extracts. The highest inhibition was recorded for August samples. Chiboub et al. [4] reported DPPH inhibition of *C. spongiosus* macerated with hexane, ethyl acetate, and methanol for 48 h at 15.6 ± 0.5%, 52.2 ± 0.0%, and 41.5 ± 0.1%, respectively, which is comparable to our results. Moreover, Yalçın et al. [37] extracted *C. spongiosus f. verticillatum* with methanol and ethanol solutions (100%, 80%, and 70%) for 120 min using UAE. The extracts showed antioxidant potential through 2,2-azino-bis(3-ethylbenzothiazoline-6-sulfonate) (ABTS) radical scavenging and cupric ion reduction. Pinteus et al. [6] showed that *C. spongiosus* harvested from the Peniche coast, and extracted with dichloromethane, has the ability to scavenge peroxyl and DPPH radicals. The antioxidant activity results mentioned above cannot be compared with our results because the authors used a different methodology or expressed their results in different units [38].

### 2.6. Non-Target Screening of Non-Volatile Compounds in Ethanol Extract

The ethanolic extract of the freeze-dried algal sample with the best antioxidant potential according to ORAC and FRAP assay (May, MAE) was analysed using UHPLC-ESI–HRMS. The major compounds were selected according to signal intensity (peak area in counts). From the extracted ion chromatograms (XICs) in positive ion mode, these compounds were identified based on the given elemental composition and MS/MS spectra with confidence level 2 (probable structure) and 3 (possible structure) [39]. Seventeen fatty acid derivatives, nine pigments and derivatives, and two steroid derivatives were found (Table 4).

Of the seventeen fatty acid derivatives, eight were fatty acid amides (FAAs) (compounds **7**–**10**, **13**, **19**, **24**), eight were glycerolipids (GLs) (compounds **3**–**6**, **11**, **12**, **14**, **18**), and one was a fatty acid (FA) (compound **9**) (Table 4). The FAAs were dominant.

Three C18 FAAs, derivatives of linoleic acid (linoleamide, no. **9**), oleic acid (oleamide, no. **13**), and stearic acid (stearamide, no. **17**), two C16 FAAs, derivatives of palmitic acid (palmitamide, no. **10**) and palmitoleic acid (palmitoleamide, no. **8**), and one C14, derivative of myristic acid (myristamide, no. **7**), C20 derivative of (11*Z*)-icos-11-enoic acid (gondamide, no. **19**), and C22 FAA derivative of erucic acid (erucamide, no. **24**) were detected. In our previous studies, we detected oleamide, erucamide, and palmitamide in the green alga *Dasycladus vermicularis* [40] and two others, palmitoleamide and linoleamide, in the brown algae *Ericaria crinita* and *Ericaria amentacea* [28]. The FAAs detected and identified in *C. spongiosus* belong to the group of primary fatty acid amides (PFAMs) with the structure R-CO-NH_2_, where R represents a long-chain FA [41], and are bioactive signalling molecules. In general, PFFAs have a broad therapeutic spectrum including anticancer, antibiotic, anthelmintic and antidiabetic [42]. They play an important role in the nervous system of the mammals because they have the potential to bind to the receptors of many drugs, indicating effects on locomotion, angiogenesis, and sleep [43]. It has been reported that PFAMs have a great anticancer activity as they effect cell proliferation [44]. Oleamide is the most studied PFFA and has great potential against Alzheimer disease [45]. Certain plant essential oils are rich in PFFAs because they act as self-defence agents in plants [42], but data on the occurrence of PFAAs in marine algae are lacking.

Among the GLs were four glycoglycerolipids, more specifically monogalactosylmonoacylglycerols (MGMGs) (compounds **3**–**6**) and four monoacylglycerols (MG) (compounds **11**, **12**, **14**, **18**). All eukaryotes and prokaryotes can synthesise GLs which form membrane lipids [46]. Marine algae are known to be a great source of lipid molecules with wide-ranging bioactive properties [47]. Glycoglycerolipids are known for their antiviral, antitumor, antimicrobial, anti-inflammatory, and antioxidant properties [48]. They can be classified into three groups based on the nature of the glycosyl and acyl structures: monogalactosyl diacylglycerols (MGDG), digalactosyl diacylglycerols (DGDG), and sulfoquinovosyldiacyl glycerols (SQDG) [49]. MGMG is a byproduct of the degradation of accumulated MGDG in the outer membrane of chloroplasts [50].

In the group of pigments, one chlorophyll *a* derivative (pheophorbide *a*, compound **21**), one demetallized chlorophyll *b* (pheophytin *b*, compound **27**), three pheophorbide derivatives (compounds **22**, **23**, **25**), one xanthophyll and two derivatives (compounds **15**, **16**, **28**), and one monoterpene lactone (loliolide, compound **1**) were detected (Table 4). The pigments are known antioxidants [51,52]. Pheophorbide *a*, a derivative of chlorophyll *a*, has been extracted from marine algae and has shown great antitumor activity [53]. Among the pigments, it was the most abundant in the extract (Table 4), which contributes to the antioxidant activity of the extract. Cho et al. [54] studied the green alga *Ulva prolifera* (formerly *Enteromorpha prolifera*) and its ethanolic extract, which showed high antioxidant activities. The later subfraction with chloroform showed the strongest antioxidant activity of all subfractions, and further spectroscopic analysis revealed the pheophorbide *a* as the major compound.

Fucoxanthin, the major xanthophyll in brown algae [55,56], and xantophyll derivatives, halocynthiaxanthin acetate and loroxanthin decenoate, were detected. El Hattab et al. [57] reported fucoxanthin in *C. spongiosus f. verticillatus*. Halocynthiaxanthin acetate and loroxanthin decenoate are more abundant in green algae [58]. Xanthophylls exhibit diverse bioactive properties, such as antioxidant, antitumor, and anti-inflammatory activities [55,56,59]. The antioxidant activity of the extract might be related to the high amounts of fucoxanthin (Table 4). Ibrahim et al. [60] evaluated the potent antimicrobial activity against Gram-positive and Gram-negative bacteria and fungi and the strong antioxidant activity of fucoxanthin extracted from *Dictyota fasciola* (formerly *Dilophys fasciola*). Fucoxanthin is one of the most studied xanthophylls against cancer cells because of its antiproliferative effect [59]. Halocynthiaxanthin is a metabolite of fucoxanthin and has shown even stronger cytotoxic results than fucoxanthin when tested on human neuroblastoma cells [61]. Sansone et al. [62] studied the biological activity of the marine green alga *Tetraselmis suecica*, whose ethanolic extract contained a high proportion of xhantophylls, among which loraxanthin esters were present. The extract showed potent antioxidant activity and antitumor activity against human lung cancer line (A549).

Loliolide, the only monoterpernoid lactone detected in our study, was previously detected in *C. spongiosus f. verticillatus* by El Hattab et al. [57]. It has also been detected in the genus *Ericaria* [28], *D. vermicularis* [40], and the microalga *Tetradesmus obliquus* [63]. Silva et al. [64] demonstrated the neuroprotective, anti-inflammatory, and antioxidant effects of loliolide in an extract of *Codium tomentosum*.

## 3. Materials and Methods

### 3.1. Macroalga Samples

Between May and August 2021, the samples of the brown alga *C. spongiosus* (Hudson) C. Agardh, 1817 were collected. The sampling took place in the Adriatic Sea, off the coast of the island of Čiovo (43.493373° N, 16.272519° E). Each sample was collected at a depth of 20 to 120 cm from the same lagoon. The sea temperature was measured during each sampling with a YSI Pro2030 probe (Yellow Springs, OH, USA) and increased from 20.1 °C in May to 28.1 °C in August. Alga species was determined according to its morphological attributes by marine botanist. For the determination of VOCs, the samples were dried in the shade at room temperature for 10 days, while samples for the determination of non-volatile compounds were freeze-dried (FreeZone 2.5, Labconco, Kansas City, MO, USA) prior the extraction. Algal samples were extracted in 50% ethanol using MAE in the advanced microwave extraction system (ETHOS X, Milestone Srl, Sorisole, Italy) and UAE based on the prior research [9]. The samples were pulverised and mixed with 50% ethanol at a 1:10 (*w*/*v*) algae to solvent ratio and extracted for: (1) 15 min at 200 W and 60 °C for MAE, and (2) 60 min at 40 kHz frequency and 60 °C in an ultrasonic bath for UAE.

### 3.2. Headspace Solid-Phase Microextraction (HS-SPME)

HS-SPME was performed on the PAL Auto Sampler System (PAL RSI 85, CTC Analytics AG, Schlieren, Switzerland) using two SPME fibres of different polarity: PDMS/DVB (polydimethylsiloxane/divinylbenzene) or VB/CAR/PDMS (divinylbenzene/carboxene/polydimethylsiloxane) (Agilent Technologies, Palo Alto, Santa Clara, CA, USA). Both fibres were conditioned before analysis according to the manufacturer’s instructions. Samples (1 g) were placed in 20 mL glass vials sealed with a stainless steel lid with polytetrafluorethylene (PTFE)/silicon septum. First, the sample was equilibrated at a temperature of 60 °C for 15 min, after which the extraction was continued for 45 min. Subsequent thermal desorption lasted 6 min in the inlet set at 250 °C, from where the compounds were passed directly into the GC column.

### 3.3. Hydrodistillation (HD)

Approximately 50 g of fresh and 30 g of air-dried algal material were hydrodistilled in a modified Clevenger apparatus containing 3 mL of organic solvent trap (pentane and diethyl ether, 1:2 *v*/*v*, both from Kemika, Zagreb, Croatia). After 2.5 h of hydrodistillation, the VOC-containing organic solvent trap was collected and concentrated to a final volume of 0.2 mL under slow nitrogen flow. GC-MS analysis was performed using 2 µL of the sample.

### 3.4. Gas Chromatography–Mass Spectrometry Analysis (GC–MS)

GC-MS analysis was performed using an Agilent Technologies (Palo Alto, CA, USA) model 8890A gas chromatograph, coupled to a model 5977E mass selective detector. The compounds were separated on a HP -5MS column (Agilent Technologies, Santa Clara, CA, USA) 30 m × 0.25 mm with a nonpolar stationary phase (5% diphenyl/95% dimethylpolysiloxane) and a film thickness of 0.25 μm. The following operating conditions were used for the gas chromatograph: 250 °C injector temperature; 300 °C detector temperature; column temperature program: 2 min isothermal at 70 °C, followed by a temperature gradient from 70 °C to 200 °C at 3 °C/min and a further retention of 15 min. The split ratio was 1:50; the carrier gas was helium with a flow rate of 1.0 mL/min; the MSD (EI mode) was operated at 70 eV; the mass range was set to 30 to 300 amu. The compounds were identified by comparing their retention indices (RI), which were based on the retention times of *n*-alkanes (C_9_–C_25_), with those reported in the literature (National Institute of Standards and Technology) and their mass spectra with those from the Wiley 9 (Wiley, New York, NY, USA) and NIST 17 (D-Gaithersburg) mass spectral libraries. Percent composition was calculated using the normalisation method (without correction factors). Analyses were performed in duplicate and expressed as the average percentage of peak area.

### 3.5. Ultra-High Performance Liquid Chromatography-High-Resolution Mass Spectrometry (UHPLC-ESI-HRMS) of Ethanol Extract

The UHPLC-ESI-HRMS analyses were performed using the ExionLC AD UHPLC system (AB Sciex, Concord, ON, Canada) equipped with the following ExionLC modules: AD Controller, solvent delivery system (AD Pump and AD Degasser), AD Autosampler and AD Column oven, connected to quadrupole time-of-flight (Q-TOF) mass spectrometer TripleTOF 6600+ (AB Sciex, Concord, ON, Canada) with duospray ion source. Chromatographic separations of the compounds were performed using the Acquity UPLC BEH Phenyl-Hexyl analytical column (Waters, Milford, MA, USA) 2.1 mm × 100 mm with a particle size of 1.7 µm. A continuous flow rate of 0.4 mL/min was set to pump the mobile phases, water (A) and acetonitrile (B), both containing 0.1% formic acid. The oven temperature was set at 30 °C. Elution started isocratically with 2% B (0.6 min), followed by a gradient program: 0.6–18.5 min (linear B gradient to 100%), 18.5–25 min (100% B). Electrospray ionisation was set in positive mode (ESI^+^) with collision-induced dissociation (CID) in information-dependent acquisition mode (IDA) for MS/MS mass spectra acquisition. In our previous work, description of the parameters in details can be found [40]. ACD/Spectrus Processor 2021.1.0 software (ACD/Labs, Toronto, ON, Canada) was used to process the mass spectrometer data. Based on the mass spectra and the given elemental compositions of the compounds, and in conjunction with the results of the search in the ChEBI, Lipid Maps, and MassBank databases, the identification of the compounds was proposed.

### 3.6. Antioxidant Activity of Extracts

Three different methods were used to assess in vitro antioxidant activity of crude algal extracts. The oxygen radical absorbance capacity (ORAC) and 2,2-diphenyl-1-picrylhydrazyl radical scavenging ability (DPPH) methods are based on hydrogen atom transfer, while ferric reducing/antioxidant power (FRAP) method is based on electron transfer [11,65,66].

Previously reported methods [11,65] were used to measure the reducing activity as FRAP. The results were expressed using the Trolox standard calibration curve (*y* = 0.0013*x*, *R*^2^ = 0.99) as micromoles of Trolox equivalents (µM TE). The inhibition of the free peroxyl radicals, measured as ORAC [11], was measured for the samples in 1:100 dilution, and the results were expressed using the Trolox standard calibration curve (*y* = 22.842*x* + 26.473, *R*^2^ = 0.99) as µM TE. DPPH radical inhibition was expressed as a percentage of inhibition and measured according to the previously reported methods [11,67].

### 3.7. Statistical Analyses

The relationship between the dominant volatiles (>2%) of fresh and dried *C. spongiosus* samples was determined via principal component analysis (PCA) using the software STATISTICA^®^ (version 13, StatSoft Inc., Tulsa, OK, USA). Before analyses, the data (average percentage of peak areas of the dominant volatiles) were log-transformed [9].

## 4. Conclusions

HS-SPME showed that aliphatic compounds dominated in all samples. Hydrocarbons were more than twice as high in the dry samples (except in May). Aliphatic alcohols were present in large amounts and were more abundant in the fresh samples. Hexanal, heptanal, nonanal, and tridecanal formed the majority of aldehydes. Aliphatic ketones were more abundant in the fresh samples. Benzene derivatives were predominant in the fresh samples of May and August. Mono- and sesquiterpenes (mainly (*E*)-verbenol and *p*-cymen-8-ol) were the most abundant in the May dry sample.

HD obtained four groups of compounds: aliphatic compounds (e.g., heptadecane, pentadecanal, (*E*)-heptadec-8-ene and (*Z*)-heptadec-3-ene with the highest percentage in August), sesquiterpenes (germacrene D, epi-bicyclosesquiphellandrene, gleenol), diterpenes (phytol, pachydictyol A, (*E*)-geranyl geraniol, cembra-4,7,11,15-tetraen-3-ol) and others. Except for the July fresh sample, terpenes were the most abundant. The lowest percentage of diterpenes was always in fresh samples and the highest in dry samples.

Seasonal variations in antioxidant activity of *C. spongiosus* extracts obtained with MAE and UAE were confirmed. All MAE extracts showed higher activity. The highest peroxyl radical inhibition and reducing activity was observed in the May samples. Therefore, the ethanolic extract from May was analysed by UHPLC-ESI-HRMS. A total of 17 fatty acid derivatives, 9 pigments and derivatives, and 2 steroid derivatives were found. The highest content of pheophorbide *a* and fucoxanthin and the presence of other pigment derivatives could be related to the observed antioxidant activity.

Further research should focus on exploring the seasonal variations of the non-volatile compounds of this alga and the potential use of the extracts obtained.

## Figures and Tables

**Figure 1 marinedrugs-21-00415-f001:**
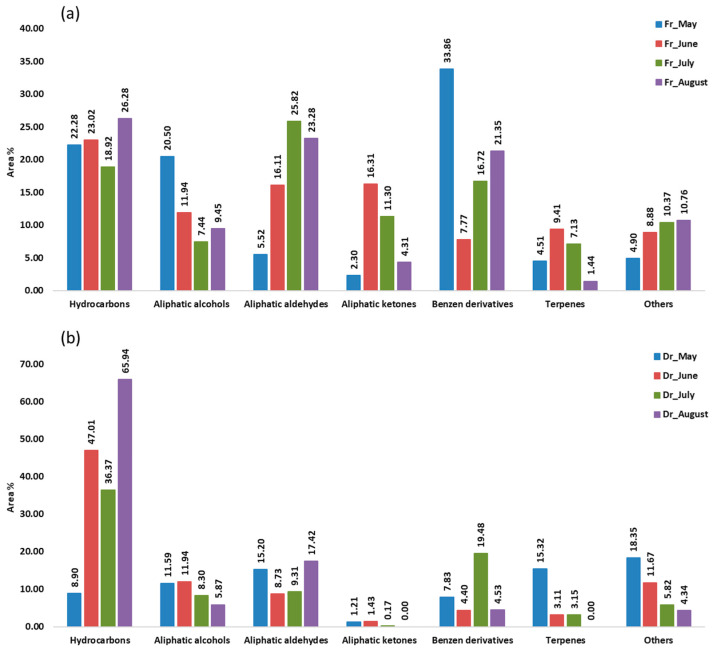
The volatile organic compounds (VOCs) from *Cladostephus spongiosus* sorted by structural groups obtained by headspace solid-phase microextraction (HS-SPME) with divinylbenzene/carboxene/polydimethylsiloxane fibre (f1) and analysed by gas chromatography–mass spectrometry (GC-MS): (**a**) fresh (Fr) samples; (**b**) dry (Dr) samples.

**Figure 2 marinedrugs-21-00415-f002:**
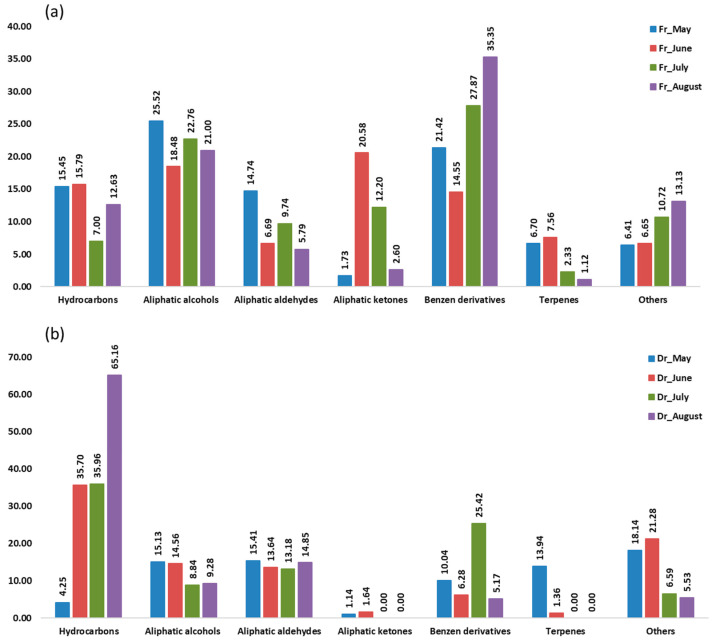
The volatile organic compounds (VOCs) of *Cladostephus spongiosus* sorted by structural groups obtained via headspace solid-phase microextraction (HS-SPME) with polydimethylsiloxane/divinylbenzene fibre (f2) and analysed via gas chromatography–mass spectrometry (GC-MS): (**a**) fresh (Fr) samples; (**b**) dry (Dr) samples.

**Figure 3 marinedrugs-21-00415-f003:**
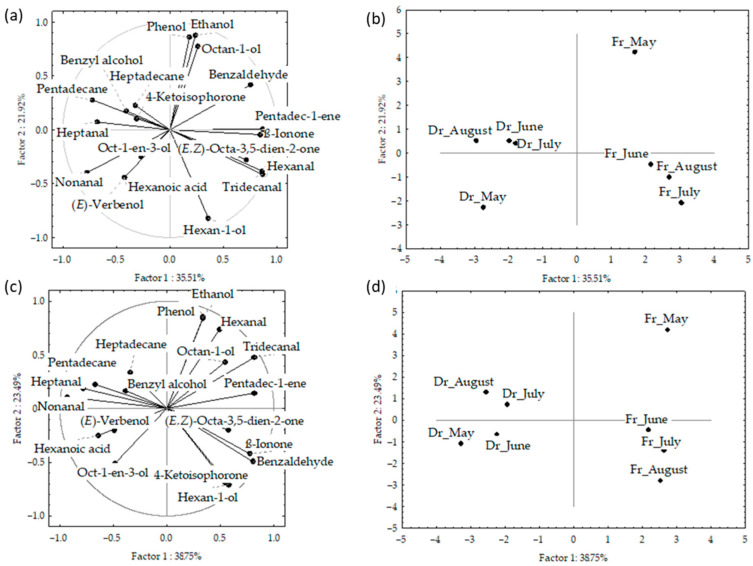
Correlation loadings (**a**,**c**) and score plots (**b**,**d**) of the dominant compounds from the headspace volatiles obtained via headspace solid-phase microextraction (HS-SPME) with two fibres divinylbenzene/carboxene/polydimethylsiloxane (f1) and polydimethylsiloxane/divinylbenzene (f2) and analysed via gas chromatography–mass spectrometry (GC-MS): fresh (Fr) and dried (Dr) *Cladostephus spongiosus* samples collected from May to August.

**Figure 4 marinedrugs-21-00415-f004:**
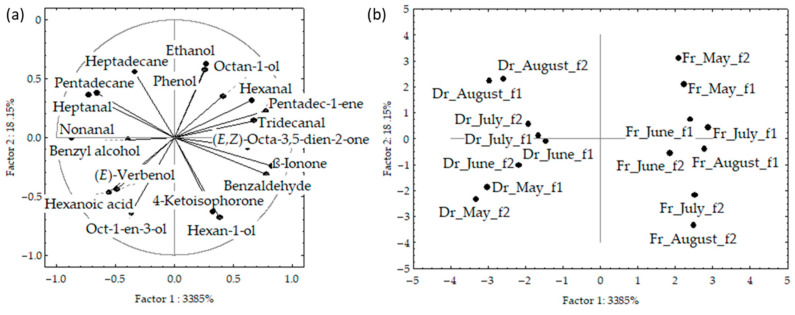
Correlation (**a**) and score plots (**b**) of the dominant compounds from headspace volatiles (two different fibres obtained via headspace solid-phase microextraction (HS-SPME) with two fibres divinylbenzene/carboxenene/polydimethylsiloxane (f1) and polydimethylsiloxane/divinylbenzene (f2)) and analysed via gas chromatography–mass spectrometry (GC-MS): fresh (Fr) and dried (Dr) *Cladostephus spongiosus* samples analysed together.

**Figure 5 marinedrugs-21-00415-f005:**
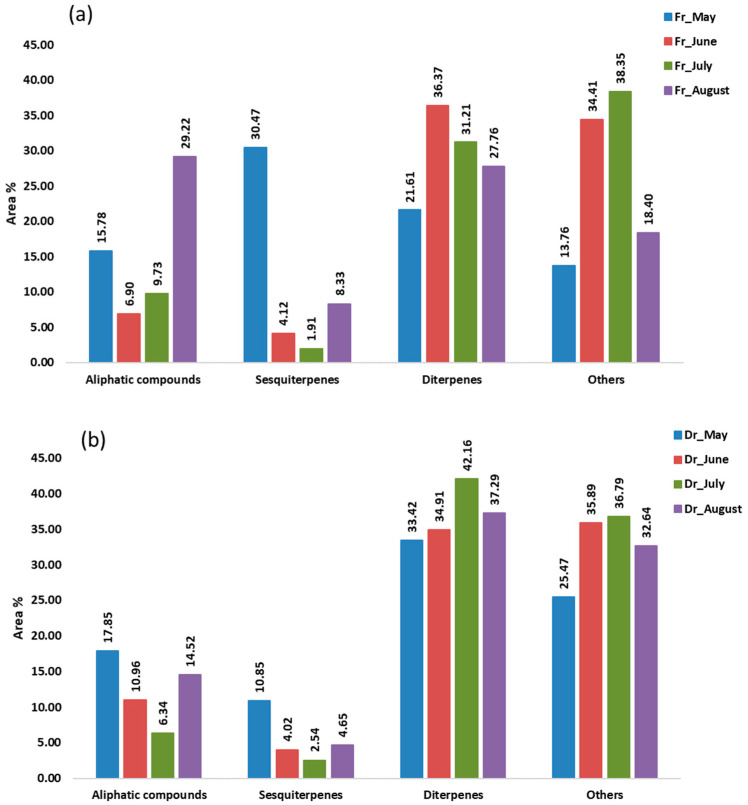
The volatile organic compounds (VOCs) from *Cladostephus spongiosus* sorted by structural groups isolated via hydrodistillation (HD) and analysed via gas chromatography–mass spectrometry (GC-MS): (**a**) fresh (Fr) samples; (**b**) dry (Dr) samples.

**Figure 6 marinedrugs-21-00415-f006:**
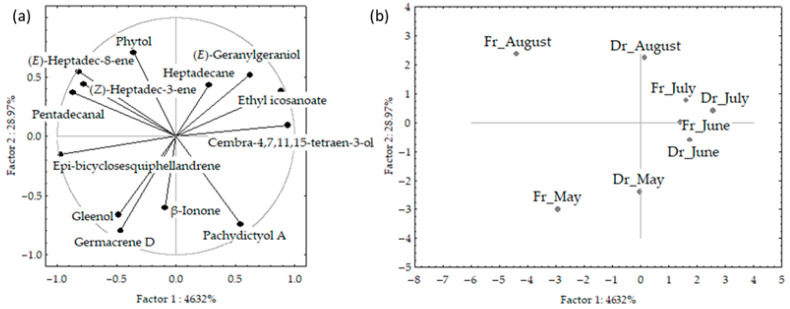
Correlation loadings (**a**) and score plot (**b**) of the dominant volatile organic compounds (VOCs) of fresh and air-dried *Cladostephus spongiosus* samples obtained via hydrodistillation and analysed via gas chromatography-mass spectrometry (GC-MS).

**Figure 7 marinedrugs-21-00415-f007:**
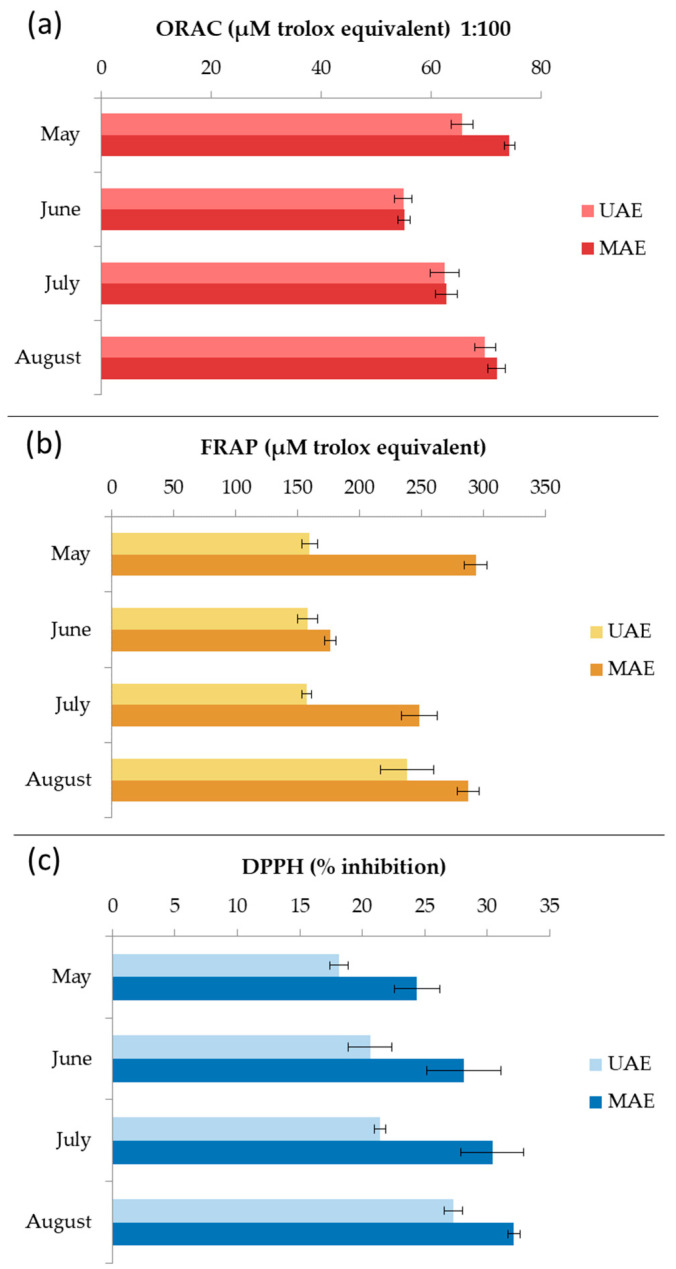
Seasonal variation of the oxygen radical absorbance capacity (ORAC) (**a**), ferric reducing/antioxidant power (FRAP) (**b**), and 2,2-diphenyl-1-picrylhydrazyl radical scavenging ability (DPPH) (**c**) results for *Cladostephus spongiosus* extracts obtained with ultrasound-assisted extraction (UAE) and microwave-assisted extraction (MAE).

**Table 1 marinedrugs-21-00415-t001:** The volatile organic compounds (VOCs) of *Cladostephus spongiosus* obtained via headspace solid-phase microextraction (HS-SPME) with divinylbenzene/carboxen/polydimethylsiloxane fibre (f1) and analysed via gas chromatography–mass spectrometry (GC-MS): fresh (Fr) samples, dry (Dr) samples.

No.	Compound	Group	RI	Area %							
				May		June		July		August	
				Fr	Dr	Fr	Dr	Fr	Dr	Fr	Dr
**HYDROCARBONS**											
19	3,5,5-Trimethylhex-2-ene	Hydrocarbon (UnSAC)	980	-	-	-	2.75	-	0.72	-	-
55	2,6,11-Trimethyldodecane	Hydrocarbon (SAC)	1282	-	-	-	-	0.64	-	3.61	-
62	Tetradecane	Hydrocarbon (SAC)	1400	-	0.58	-	0.69	-	0.62	-	-
66	Pentadec-1-ene	Hydrocarbon (UnSAC)	1495	6.39	0.49	11.35	0.99	6.22	1.65	10.6	2.78
67	Pentadecane	Hydrocarbon (SAC)	1500	9.65	6.39	7.43	41.58	3.53	28.96	5.97	43.88
73	(*E*)-Heptadec-8-ene	Hydrocarbon (UnSAC)	1690	1	-	0.87	-	-	-	-	-
74	Heptadec-1-ene	Hydrocarbon (UnSAC)	1696	0.96	0.62	1.8	-	1.46	0.92	2.19	1.62
75	Heptadecane	Hydrocarbon (SAC)	1700	4.29	0.81	1.57	1	2.07	2.1	3.91	17.66
77	(*Z*)-Heptadec-3-ene	Hydrocarbon (UnSAC)	1720	-	-	-	-	-	1.4	-	-
78	Octadec-1-ene	Hydrocarbon (UnSAC)	1791	-	-	-	-	3.1	-	-	-
81	Nonadec-1-ene	Hydrocarbon (UnSAC)	1897	-	-	-	-	1.89	-	-	-
**ALIPHATIC ALCOHOLS**											
1	Ethanol	Alcohol (SAC)	<900	11.09	0.75	1.23	2.91	2.19	1.5	2.05	2.73
5	Pent-1-en-3-ol	Alcohol (UnSAC)	<900	0.29	0.56	0.24	0.45	0.67	0.31	-	-
6	3-Methylbutan-1-ol	Alcohol (SAC)	<900	-	0.42	-	-	-	-	-	-
7	Pentan-1-ol	Alcohol (SAC)	<900	-	0.38	-	0.28	-	-	-	-
8	(*Z)*-Pent-2-en-1-ol	Alcohol (UnSAC)	<900	0.68	0.26	0.29	0.29	0.25	-	0.44	-
13	Hexan-1-ol	Alcohol (SAC)	<900	0.3	1.47	0.82	0.62	2.27	0.82	1.38	0.66
18	Heptan-1-ol	Alcohol (SAC)	975	-	1.24	-	0.58	-	-	-	-
21	Oct-1-en-3-ol	Alcohol (UnSAC)	984	1.36	3.7	1.44	3.74	0.84	3.36	4.06	0.71
28	2-Ethylhexan-1-ol	Alcohol (SAC)	1034	-	-	0.26	0.4	-	0.9	-	1.91
32	(*Z*)-Oct-3-en-1-ol	Alcohol (UnSAC)	1056	-	-	2.67	-	-	-	-	-
35	(*E*)-Oct-2-en-1-ol	Alcohol (UnSAC)	1073	-	0.58	-	0.74	-	0.66	-	-
37	Octan-1-ol	Alcohol (SAC)	1076	6.76	2.22	4.13	2.31	1.22	1.66	1.53	1.77
83	(*Z*,*Z*,*Z*)-9,12,15-Octadecatrien-1-ol	Alcohol (UnSAC)	2055	-	-	1.12	-	-	-	-	-
**ALIPHATIC ALDEHYDES**											
3	Butanal	Aldehyde (SAC)	<900	0.33	-	-	0.23	-	-	0.39	-
4	3-Methylbutanal	Aldehyde (SAC)	<900	0.31	0.18	0.39	-	0.36	-	-	-
9	Hexanal	Aldehyde (SAC)	<900	2	1.04	3.77	1.12	5.83	1.3	4.62	1.84
12	(*E*)-Hex-2-enal	Aldehyde (UnSAC)	<900	0.45	0.47	0.86	0.39	0.87	0.58	0.93	0.43
14	Heptanal	Aldehyde (SAC)	907	-	1.26	-	1.52	-	1.53	-	7.47
16	(*E*)-Hept-2-enal	Aldehyde (UnSAC)	963	-	-	0.26	0.16	0.45	0.55	-	-
26	Octanal	Aldehyde (SAC)	1007	0.57	2.73	1.06	0.69	0.58	0.89	0.64	1.37
27	(*E*,*E*)-Hepta-2,4-dienal	Aldehyde (UnSAC)	1016	-	0.29	0.35	0.49	-	0.55	-	-
34	(*E*)-Oct-2-enal	Aldehyde (UnSAC)	1064	-	0.47	0.75	0.66	0.5	0.73	2.87	-
41	Nonanal	Aldehyde (SAC)	1108	0.45	5.89	1.11	2.08	1.89	2.14	0.88	5.41
46	(*Z*)-Non-2-enal	Aldehyde (UnSAC)	1165	-	-	-	-	-	-	0.67	-
51	Decanal	Aldehyde (SAC)	1209	-	1.34	-	-	-	-	-	0.9
54	(*E*)-Dec-2-enal	Aldehyde (UnSAC)	1267	-	0.8	-	-	-	-	-	-
56	(*E*,*Z*)-Deca-2,4-dienal	Aldehyde (UnSAC)	1296	-	-	0.47	0.43	0.87	0.45	0.79	-
57	Undecanal	Aldehyde (SAC)	1311		0.74	-	0.47	-	-	-	-
58	(*E*,*E*)-Deca-2,4-dienal	Aldehyde (UnSAC)	1320	0.17	-	0.71	0.49	0.82	0.6	1.22	-
63	Dodecanal	Aldehyde (SAC)	1412	-	-	0.51	-	0.78	-	0.9	-
68	Tridecanal	Aldehyde (SAC)	1514	1.24	-	4.29	-	8.41	-	4.94	-
76	Pentadecanal	Aldehyde (SAC)	1718	-	-	1.56	-	4.47	-	4.41	-
**ALIPHATIC KETONES**											
23	Octan-3-one	Ketone (SAC)	991	-	-	0.47	-	-	-	1.07	-
24	6-Methylhept-5-en-2-one	Ketone (UnSAC)	992	-	0.94	-	0.71	-	-	-	-
39	(*E*,*Z*)-Octa-3,5-dien-2-one	Ketone (UnSAC)	1097	2.3	0.27	15.84	0.71	11.3	0.17	3.24	-
**BENZENE DERIVATIVES**											
17	Benzaldehyde	Benzene derivative (aldehyde)	970	22.92	2.39	4.78	2.35	13.06	1.88	16.82	1.74
22	Phenol	Benzene derivative (phenol)	986	4.82	-	-	-	-	1.29	-	-
29	Benzyl alcohol	Benzene derivative (alcohol)	1041	4.23	5.05	0.93	1.83	1.34	12.67	-	1.1
31	Phenylacetaldehyde	Benzene derivative (aldehyde)	1052	-	0.39	0.51	0.23	0.49	-	0.74	-
36	Acetophenone	Benzene derivative (ketone)	1074	1.17	-	0.51	-	-	-	0.91	-
61	4-(2-Methylbutan-2-yl) phenol (p-tert-Pentylphenol)	Benzene derivative (phenol)	1400	-	-	-	-	0.42	-	-	-
69	2,4-Di*tert*-butylphenol	Benzene derivative (phenol)	1518	0.72	-	1.04	-	1.42	3.64	2.88	1.68
**TERPENES**											
30	(*E*)-β-Ocimene	Terpene (monoterpene)	1044	-	-	-	-	1.25	-	-	-
40	Perillene (3-(4-methylpent-3-enyl)furan)	Terpene (monoterpenoid furan)	1105	-	0.41	-	-	-	-	-	-
44	(*E*)-Verbenol	Terpene (monoterpene alcohol)	1152	-	8.8	-	-	-	-	-	-
47	1,4-Dimethyl-3-methylidenebicyclo[2.2.1] heptan-2-one	Terpene (monoterpene ketone)	1169	-	3.84	-	-	-	-	-	-
50	*p*-Cymen-8-ol	Terpene (monoterpene alcohol)	1190	-	3.71	-	-	-	-	-	-
52	β-Cyclocitral	Terpene (monoterpene aldehyde)	1226	0.59	-	1.19	0.61	1.25	0.98	1.44	-
53	*p*-Cumic aldehyde	Terpene (monoterpene aldehyde)	1246	-	1.46	-	-	-	-	-	-
59	β-Bourbonene	Terpene (sesquiterpene)	1389	-	-	0.5	0.76	-	-	-	-
60	β-Cubebene	Terpene (sesquiterpene)	1393	-	-	0.79	-	-	-	-	-
64	Germacrene D	Terpene (sesquiterpene)	1485	2.08	-	2.62	-	-	-	-	-
70	δ-Cadinene	Terpene (sesquiterpene)	1528	-	-	0.2	-	-	-	-	-
72	Gleenol	Terpene (sesquiterpene alcohol)	1589	1.84	-	4.11	-	-	-	-	-
79	Phytane	Terpene (diterpene)	1813	-	-	-	-	3.57	1.13	-	-
80	Hexahydrofarnesyl acetone	Terpene (sesquiterpene ketone)	1850	-	0.94	-	1.75	1.06	1.04	-	-
**OTHERS**											
2	Dimethyl sulfide	Other (sulfide)	<900	-	-	0.8	-	0.47	0.3	0.77	0.68
10	3-Methylbutanoic acid	Other (carboxylic acid)	<900	-	0.72	-	1.25	-	-	-	-
11	2-Methylbutanoic acid	Other (carboxylic acid)	<900	-	0.23	-	0.62	-	-	-	-
15	2-Iodopentane	Other	932	-	-	0.5	-	0.91	0.57	0.63	0.56
20	Hexanoic acid	Other (carboxylic adic)	981	-	4.75	-	-	-	-	-	-
25	2-Pentylfuran	Other (furan)	996	0.71	0.55	0.35	0.47	0.49	0.78	-	0.68
33	γ-Caprolactone (5-ethyloxolan-2-one)	Other (γ-lactone)	1062	-	0.38	-	0.59	-	-	-	-
38	Heptanoic acid	Other (carboxylic adic)	1079	-	2.16	-	1.17	-	-	-	-
42	2,6-Dimethylcyclohexanol	Other	1114	-	1.96	0.55	3.67	0.87	1.2	0.86	1.05
43	4-Ketoisophorone	Other (C_13_-norisoprenoid)	1150	-	-	-	0.8	-	-	-	-
45	Dictyopterene D’ (6-[(1*Z*)-butenyl]-1,4-cycloheptadiene])	Other (dictyopterene)	1158	0.81	-	2.94	-	2.55	-	2.2	-
48	Dictyopterene C’ ([6-butyl-1,4-cycloheptadiene])	Other (dictyopterene)	1175	-	-	0.49	-	1.33	-	0.89	-
49	Octanoic acid	Other (carboxylic acid)	1177	-	0.59	-	-	-	-	-	-
65	β-Ionone	Other (C_13_-norisoprenoid)	1489	3.39	1.71	2.75	2.26	3.74	2.27	5.41	1.37
71	Dihydroactinidiolide (4,4,7a-trimethyl-6,7-dihydro-5H-1-benzofuran-2-one)	Other (benzofuran)	1532	-	1.46	-	0.81	-	0.69	-	-
82	Methyl (all *Z*) 5,8,11,14- eicosatetraenoate	Other	2049	-	-	0.5	-	-	-	-	-
**Total identified=**				**93.87**	**74.56**	**93.69**	**88.69**	**97.69**	**83.5**	**96.86**	**100**

No—order of the compounds elution on HP-5MS column; RI—retention index; SAC—Saturated aliphatic compound; UnSAC—Unsaturated aliphatic compound.

**Table 2 marinedrugs-21-00415-t002:** The volatile organic compounds (VOCs) from *Cladostephus spongiosus* obtained via headspace solid-phase microextraction (HS-SPME) with polydimethylsiloxane/divinylbenzene fibre (f2) and analysed via gas chromatography–mass spectrometry (GC-MS): fresh (Fr) samples; dry (Dr) samples.

No.	Compound	Group	RI	Area %							
				May		June		July		August	
				Fr	Dr	Fr	Dr	Fr	Dr	Fr	Dr
**HYDROCARBONS**											
21	3,5,5-Trimethylhex-2-ene	Hydrocarbon (UnSAC)	979	-	-	0.68	-	0.42	0.67	0.58	-
53	2,6,11-Trimethyldodecane	Hydrocarbon (SAC)	1282	-	-	-	-	0.04	-	1.68	-
62	Pentadec-1-ene	Hydrocarbon (UnSAC)	1495	4.49	-	6.36	0.81	2.45	2.07	4.21	1.84
63	Pentadecane	Hydrocarbon (SAC)	1500	5.44	3.5	6.83	33.83	2.48	31.06	3.75	35.68
69	(*E*)-Heptadec-8-ene	Hydrocarbon (UnSAC)	1690	0.66	-	0.49	-	-	-	-	-
70	Heptadec-1-ene	Hydrocarbon (UnSAC)	1696	1.28	-	0.45	-	-	-	0.73	1.93
71	Heptadecane	Hydrocarbon (SAC)	1700	3.59	0.75	0.98	1.06	1.05	2.15	1.68	25.71
73	Octadec-1-ene	Hydrocarbon (UnSAC)	1791	-	-	-	-	0.58	-	-	-
**ALIPHATIC ALCOHOLS**											
1	Ethanol	Alcohol (SAC)	<900	17.15	1.12	0.93	2.67	3.09	3.38	2.38	4.89
4	Isobutanol	Alcohol (SAC)	<900	-	0.29	-	-	-	-	-	-
6	Pent-1-en-3-ol	Alcohol (UnSAC)	<900	-	0.77	-	0.89	-	1.03	-	0.83
8	3-Methylbutan-1-ol	Alcohol (SAC)	<900	-	0.57	0.46	0.45	0.96	-	0.85	-
9	Pentan-1-ol	Alcohol (SAC)	<900	-	0.59	0.97	0.5	1.62	-	1.9	-
10	(*Z*)-Pent-2-en-1-ol	Alcohol (UnSAC)	<900	1.36	0.54	0.37	0.65	0.42	0.56	0.59	-
15	Hexan-1-ol	Alcohol (SAC)	<900	-	3.21	4.49	1.38	12.88	-	10.34	0.98
20	Heptane-1-ol	Alcohol (SAC)	975	-	1.54	0.68	0.68	0.59	-	-	-
23	Oct-1-en-3-ol	Alcohol (UnSAC)	984	1.33	4.56	1.88	5.08	1.48	3.09	3.63	1.06
30	2-Ethylhexan-1-ol	Alcohol (SAC)	1035	-	-	-	0.39	-	-	-	1.52
34	(*Z*)-Oct-3-en-1-ol	Alcohol (UnSAC)	1056	-	-	3.98	-	0.36	-	-	-
38	Octan-1-ol	Alcohol (SAC)	1076	4.94	1.93	4.16	1.88	1.35	0.78	1.32	-
76	(*Z*,*Z*,*Z*)-9,12,15-Octadecatrien-1-ol	Alcohol (UnSAC)	2055	0.74	-	0.56	-	-	-	-	-
**ALIPHATIC ALDEHYDES**											
3	Butanal	Aldehyde (SAC)	<900	-	-	0.31	0.54	0.26	-	0.45	-
5	3-Methylbutanal	Aldehyde (SAC)	<900	0.35	-	0.38	-	0.5	-	0.29	-
7	Pentanal	Aldehyde (SAC)	<900	-	-	0.33	0.12	0.74	-	0.28	0.59
11	Hexanal	Aldehyde (SAC)	<900	5.65	1.64	1.82	2.01	3.69	2.7	2.27	2.83
14	(*E*)-Hex-2-enal	Aldehyde (UnSAC)	<900	1.37	0.75	0.47	1.21	0.59	0.43	-	-
16	Heptanal	Aldehyde (SAC)	907	-	1.87	-	3.2	-	2.27	-	6.64
18	(*E*)-Hept-2-enal	Aldehyde (UnSAC)	963	-	-	-	0.58	-	0.84	-	-
28	Octanal	Aldehyde (SAC)	1007	0.81	3.03	0.41	1.29	-	1.8	-	1
29	(*E*,*E*)-Hepta-2,4-dienal	Aldehyde (UnSAC)	1016	0.9	-	-	0.82	-	1.07	-	-
36	(*E*)-Oct-2-enal	Aldehyde (UnSAC)	1064	0.56	0.44	0.55	0.68	0.33	-	0.85	-
41	Nonanal	Aldehyde (SAC)	1108	1.12	5.35	0.6	2.48	1.01	4.07	0.32	3.79
46	(*Z*)-Non-2-enal	Aldehyde (UnSAC)	1166	-	-	-	-	-	-	0.17	-
49	Decanal	Aldehyde (SAC)	1209	-	0.78	-	-	-	-	-	-
52	(*E*)-Dec-2-enal	Aldehyde (UnSAC)	1267	-	0.49	-	-	-	-	-	-
54	(*E*,*Z*)-Deca-2,4-dienal	Aldehyde (UnSAC)	1296	0.57	-	-	-	0.36	-	-	-
55	Undecanal	Aldehyde (SAC)	1311	-	1.06	-	0.7	-	-	-	-
56	(*E*,*E*)-Deca-2,4-dienal	Aldehyde (UnSAC)	1320	-	-	0.31	-	-	-	-	-
64	Tridecanal	Aldehyde (SAC)	1514	2.24	-	1	-	1.34	-	0.45	-
72	Pentadecanal	Aldehyde (SAC)	1718	1.19	-	0.52	-	0.92	-	0.71	-
**ALIPHATIC KETONES**											
25	Octan-3-one	Ketone (SAC)	991	-	-	0.59	-	0.64	-	0.97	-
26	6-Methylhept-5-en-2-one	Ketone (UnSAC)	992	-	1.14	-	1.01	-	-	-	-
32	(*E*)-3-Octen-2-one	Ketone (UnSAC)	1045	-	-	-	-	0.78	-	-	-
40	(*E*,*Z*)-Octa-3,5-dien-2-one	Ketone (UnSAC)	1098	1.73	-	2-	0.64	10.78	-	1.63	-
**BENZENE DERIVATIVES**											
19	Benzaldehyde	Benzene derivative (aldehyde)	970	10.45	2.6	11.8	3.01	23.88	1.99	31.62	1.64
24	Phenol	Benzene derivative (phenol)	986	5.32	-	-	-	-	1.38	-	-
31	Benzyl alcohol	Benzene derivative (alcohol)	1042	4.44	6.67	1.95	2.8	2.34	15.6	3.19	2.1
33	Phenylacetaldehyde	Benzene derivative (aldehyde)	1051	-	0.77	0.39	0.46	0.34	2.73	-	-
37	Acetophenone	Benzene derivative (ketone)	1073	1.2	-	0.41	-	0.33	-	0.54	-
59	4-(2-Methylbutan-2-yl)phenol (*p*-tert-Pentylphenol)	Benzene derivative (phenol)	1400	-	-	-	-	0.34	-	-	-
65	2,4-Di*tert*-butylphenol	Benzene derivative (phenol	1518	-	-	-	-	0.64	3.73	-	1.43
**TERPENES**											
44	(*E*)-Verbenol	Terpene (monoterpene alcohol)	1152	-	8.82	-	-	-	-	-	-
48	*p*-Cymen-8-ol	Terpene (monoterpene alcohol)	1190	-	3.18	-	-	-	-	-	-
50	β-Cyclocitral	Terpene (monoterpene aldehyde)	1225	0.4	1	1.28	-	1.38	-	1.12	-
51	*p*-Cumic aldehyde	Terpene (monoterpene aldehyde)	1246	-	0.94	-	-	-	-	-	-
57	β-Bourbonene	Terpene (sesquiterpene)	1388	-	-	0.29	-	-	-	-	-
58	β-Cubebene	Terpene (sesquiterpene)	1393	0.58	-	0.47	-	-	-	-	-
60	Germacrene D	Terpene (sesquiterpene)	1485	3.72	-	2.03	-	-	-	-	-
66	δ-Cadinene	Terpene (sesquiterpene)	1530	-	-	0.33	-	-	-	-	-
68	Gleenol	Terpene (sesquiterpene alcohol)	1589	2	-	2.86	-	-	-	-	-
74	Phytane	Terpene (diterpene)	1814	-	-	-	-	0.95	-	-	-
75	Hexahydrofarnesyl acetone	Terpene (sesquiterpene ketone)	1850	-	-	0.3	1.36	-	-	-	-
**OTHERS**											
2	Dimethyl sulfide	Other	<900	1.17	-	-	-	-	0.59	-	1.96
12	3-Methylbutanoic acid	Other (carboxylic acid)	<900	-	1.47	-	2.84	-	-	-	-
13	2-Methylbutanoic acid	Other (carboxylic acid)	<900	-	0.76	-	1.87	-	-	-	-
17	2-Iodopentane	Other	932	0.61	-	0.29	-	0.65	0.75	0.48	-
22	Hexanoic acid	Other (carboxylic acid)	982	-	6.56	-	4.8	-	-	-	-
27	2-Pentylfuran	Other (furan)	996	0.76	0.57	0.42	0.76	0.62	1.04	-	0.81
35	γ-Caprolactone (5-ethyloxolan-2-one)	Other (γ-lactone)	1062	-	0.68	-	0.95	-	-	-	-
39	Heptanoic acid	Other (carboxylic acid)	1079	-	2.11	-	1.73	-	-	-	-
42	2,6-Dimethylcyclohexan-1-ol	Other	1114	-	2.81	1.13	4.35	1.94	1.6	2.22	1.13
43	4-Ketoisophorone	Other (C_13_-norisoprenoid)	1150	-	-	0.56	1.04	2.82	-	5.23	-
45	Dictyopterene D’ (6-[(1*Z*)-butenyl]-1,4-cycloheptadiene])	Other (dictyopterene)	1158	1.34	-	1.32	-	0.84	-	0.44	-
47	Dictyopterene C’ ([6-butyl-1,4-cycloheptadiene])	Other (dictyopterene)	1175	-	-	0.22	-	0.2	-	-	-
61	β-Ionone	Other (C_13_-norisoprenoid)	1490	2.53	1.13	2.71	1.96	3.66	2.6	4.76	1.62
67	Dihydroactinidiolide (4,4,7a-trimethyl-6,7-dihydro-5H-1-benzofuran-2-one)	Other (benzofuran)	1533	-	2.04	-	0.98	-	-	-	-
**Total identified=**				**91.96**	**78.05**	**90.3**	**94.45**	**92.62**	**89.99**	**91.61**	**100**

No—order of the compounds elution on HP-5MS column; RI—retention index; SAC—Saturated aliphatic compounds; UnSAC—Unsaturated aliphatic compounds.

**Table 3 marinedrugs-21-00415-t003:** The volatile organic compounds (VOCs) from *Cladostephus spongiosus* isolated via hydrodistillation (HD) and analysed via gas chromatography–mass spectrometry (GC-MS): Fr (fresh) samples; Dr (dry) samples.

No.	Compound	Group	RI	Area %							
				May		June		July		August	
				Fr	Dr	Fr	Dr	Fr	Dr	Fr	Dr
**ALIPHATIC COMPOUNDS**											
1	(*E*)-Hex-2-enal	UnSAC (aldehyde)	<900	0.22	1.36	0.15	0.57	0.09	0.12	0.15	0.22
2	Hex-3-en-1-ol	UnSAC (alcohol)	<900	-	-	-	-	0.06	-	-	-
3	Hexan-1-ol	SAC (alcohol)	<900	-	0.15	-	0.06	0.09	-	0.07	-
5	Heptan-2-one	SAC (ketone)	<900	-	0.14	-	0.05	-	0.04	-	0.08
6	Nonane	UnSAC	900	0.18	-	-	-	-	-	-	-
7	(*Z*)-Hept-4-enal	UnSAC (aldehyde)	901	-	0.15	-	-	-	0.03	-	-
8	Heptanal	SAC (aldehyde)	903	-	0.62	-	1.13	-	0.05	-	0.09
10	(*E*)-Hept-2-enal	UnSAC (aldehyde)	961	-	0.06	-	-	-	-	-	-
13	3,5,5-Trimethyl-hex-2-ene	UnSAC	979	-	-	-	-	0.02	-	-	-
14	Oct-1-en-3-ol	UnSAC (alcohol)	982	-	0.28	0.03	0.13	0.05	0.12	-	0.04
15	Octan-2,5-dione	SAC (ketone)	986	-	0.27	-	0.2	-	-	-	-
16	6-Methylhept-5-en-2-one	UnSAC (ketone)	988	-	0.22	0.06	0.11	0.11	0.07	0.23	0.26
18	Octanal	SAC (aldehyde)	1004	-	0.24	-	0.09	-	0.03	-	0.05
19	(*E*,*E*)-Hepta-2,4-dienal	UnSAC (aldehyde)	1014	0.1	0.12	0.04	0.07	0.06	0.05	-	-
20	2-Ethylhexan-1-ol	SAC (alcohol)	1034	-	-	-	-	-	0.04	0.04	0.19
23	(*E*)-Oct-2-enal	UnSAC (aldehyde)	1064	-	0.27	-	0.1	-	0.07	-	0.09
24	3,5,5-Trimethyl-hex-2-ene	UnSAC	1069	-	-	-	-	0.05	-	0.08	-
26	(*E*,*E*)-3,5-Octadien-2-one	UnSAC (ketone)	1075	-	0.14	0.09	0.12	0.06	0.08	0.11	0.05
27	Nonan-3-one	SAC (ketone)	1087	-	-	-	-	-	-	-	0.05
29	(*E*,*E*)-Octa-3,5-dien-2-one	SAC (alcohol)	1097	0.33	0.62	0.4	0.44	0.23	0.15	0.53	0.3
30	Nonanal	SAC (alcohol)	1107	-	0.35	-	0.38	-	0.06	-	0.06
35	(*E*,*Z*)-Nona-2,6-dienal	UnSAC (aldehyde)	1158	-	0.1	-	0.05	-	0.04	-	-
36	(*Z*)-Non-2-enal	UnSAC (aldehyde)	1165	-	0.15	-	0.07	-	0.06	-	-
46	Undecanal	SAC (aldehyde)	1311	-	-	-	-	-	-	0.08	0.08
47	(*E*,*E*)-Deca-2,4-dienal	UnSAC (aldehyde)	1320	0.21	0.7	0.07	0.25	0.09	0.14	0.17	0.11
50	Tetradecane	SAC	1400	0.13	0.53	0.05	0.11	0.04	0.16	0.18	0.14
51	Dodecanal	SAC (alcohol)	1412	0.07	0.14	-	0.07	0.04	0.07	0.18	0.13
54	Dodecan-1-ol	SAC (alcohol)	1478	0.14	-	-	-	-	0.19	-	-
58	Pentadec-1-ene	UnSAC	1498	0.46	0.31	0.1	0.09	0.09	0.09	0.2	0.12
59	Pentadecane	SAC	1500	1.49	0.93	1.58	0.93	0.56	0.49	2.37	1.17
60	Tridecanal	SAC (alcohol)	1514	2.11	0.73	0.31	0.31	1.17	0.3	2.65	0.42
67	Tridecan-1-ol	SAC (alcohol)	1582	-	0.23	-	-	-	-	-	-
69	Hexadecane	SAC	1600	-	-	-	-	-	-	0.26	0.24
70	Tetradecanal	SAC (aldehyde)	1616	0.22	0.29	0.06	0.17	0.15	0.15	0.6	0.17
71	(*Z*)-Hexadec-7-ene	UnSAC	1623	-	-	-	-	-	-	0.12	-
77	Heptadec-1-ene	UnSAC	1690	0.77	0.57	0.6	0.53	0.31	0.24	0.78	0.21
78	(*E*)-Heptadec-8-ene	UnSAC	1696	1.08	0.36	0.41	0.26	0.59	0.37	3.96	1.5
79	Heptadecane	SAC	1700	0.45	0.61	0.82	0.55	1.32	0.9	-	3.79
80	(*Z*)-Heptadec-3-ene	UnSAC	1704	0.53	-	-	-	-	-	5.31	-
81	Pentadecanal	SAC (aldehyde)	1718	2.42	1.47	0.44	0.63	1.87	0.47	6.61	1.61
82	Pentadecan-1-ol	SAC (alcohol)	1778	-	0.6	-	0.32	-	-	0.57	-
83	Octadec-1-ene	UnSAC	1786	0.62	0.29	0.14	0.18	-	-	-	0.12
85	(*Z*)-Hexadec-9-enal	UnSAC (aldehyde)	1796	0.11	-	-	-	0.06	-	-	-
87	Hexadecanal	SAC (aldehyde)	1820	-	-	0.08	0.12	-	-	-	0.11
89	(*Z*)-Hexadec-9-en-1-ol	UnSAC (alcohol)	1866	0.13	0.94	-	0.23	-	-	-	0.14
91	Hexadecan-1-ol	SAC (alcohol)	1885	0.22	-	0.09	-	0.14	0.38	0.4	0.52
92	Nonadec-1-ene	UnSAC	1897	2.76	0.83	0.32	0.29	1.4	0.34	2.47	0.49
93	Nonadecane	SAC	1900	-	-	-	-	-	-	0.17	-
98	Eicosane	SAC	2000	0.15	1.72	-	0.82	-	-	0.24	0.9
99	Octadecanal	SAC (adehyde)	2024	0.44	0.44	0.14	0.67	0.12	0.19	-	0.24
104	(*Z*)-Octadec-9-en-1-ol	UnSAC (alcohol)	2061	0.32	0.55	0.85	0.87	0.84	0.86	0.46	0.55
105	(*Z*,*Z*)-Octadeca-3,13-dien-1-ol	UnSAC (alcohol)	2068	-	0.41	-	-	0.05	-	-	0.31
106	Octadecan-1-ol	SAC (alcohol)	2084	0.08	-	0.06	-	0.07	-	0.23	-
**SESQUITERPENES**											
48	β-Bourbonene	Sesquiterpene	1388	0.24	0.14	0.04	-	-	-	-	-
49	β-Cubebene	Sesquiterpene	1393	1.7	0.84	0.16	0.16	-	-	0.11	-
55	Germacrene D	Sesquiterpene	1485	9.63	3.32	0.8	0.6	0.14	0.12	0.23	-
57	Epi-bicyclosesquiphellandrene	Sesquiterpene	1495	4.49	2.06	1.04	0.72	0.88	0.77	4.37	1.7
61	γ-Cadinene	Sesquiterpene	1519	0.59	0.27	0.11	-	0.1	0.12	0.56	0.32
62	δ-Cadinene	Sesquiterpene	1528	0.13	0.45	-	-	-	-	-	-
63	Zonarene	Sesquiterpene	1530	0.64	-	0.13	-	-	-	-	-
64	Cadina-4,9-diene	Sesquiterpene	1557	-	-	-	-	0.06	-	-	-
66	Germacrene-4-ol	Sesquiterpene (alcohol)	1580	0.22	-	-	-	0.05	-	0.34	-
68	Gleenol	Sesquiterpene (alcohol)	1589	11.58	0.94	1.46	0.44	0.2	0.17	0.52	-
73	τ-Cadinol	Sesquiterpene (alcohol)	1647	0.39	-	-	-	-	-	-	-
74	δ-Cadinol	Sesquiterpene (alcohol)	1651	0.21	-	-	-	-	-	0.67	0.5
75	α-Cadinol	Sesquiterpene (alcohol)	1659	0.26	-	-	-	-	-	-	-
88	Hexahydrofarnesyl acetone	Sesquiterpene (ketone)	1850	0.38	2.34	0.39	1.77	0.49	1.24	1.54	1.83
94	(*E*,*E*)-Farnesyl acetone	Sesquiterpene (ketone)	1923	-	0.49	-	0.34	-	0.13	-	0.3
**DITERPENES**											
84	Phyt-1-ene	Diterpene	1791	-	-	-	-	0.49	-	-	-
86	Phytane	Diterpene	1813	-	-	-	-	0.65	0.21	0.36	-
101	Thunbergol (Cembra-2,7,11-trien-4-ol)	Diterpene	2045	2.32	2.47	2.58	2.17	1.99	2.74	0.81	0.99
95	Isophytol	Diterpene (alcohol)	1953	0.27	0.32	0.06	0.17	0.07	0.18	0.36	0.21
110	Phytol	Diterpene (alcohol)	2116	4.89	7.04	9.3	5.52	4.42	6.48	14.93	18.73
111	Pachydictyol A	Diterpene (alcohol)	2127	7.81	12.44	8.9	10.21	6.68	9.73	3.18	4.47
113	(*E*)-Geranylgeraniol	Diterpene (alcohol)	2208	1.4	1.79	3.23	2.84	5.08	6.77	2.79	3.88
114	Cembra-4,7,11,15-tetraen-3-ol	Diterpene (alcohol)	2230	4.92	9.37	12.3	13.99	11.83	16.04	5.34	9.02
**OTHERS**											
9	2-Iodopentane	Other	930	-	-	-	0.03	-	-	-	0.04
12	Dimethyl disulfide	Other	978	-	0.11	-	0.05	-	0.03	-	-
21	2,2,6-Trimethylcyclohexanone	Other	1041	-	0.11	-	-	-	-	-	-
28	1-Methylsulfanylpentan-3-one	Other	1091	0.19	-	0.05	0.1	0.03	-	-	-
31	2,6-Dimethylcyclohexan-1-ol	Other	1114	-	0.25	0.06	0.17	0.03	0.11	0.09	0.1
37	Benzyl methyl sulfide	Other	1170	-	0.12	-	-	-	-	-	-
43	Benzothiazole	Other	1228	-	-	-	-	-	-	0.15	-
76	Cyclotetradecane	Other	1681	0.23	0.81	0.08	0.41	0.12	0.3	0.62	0.34
4	*m*-Xylene	Other (benzene derivative)	<900	0.09	0.26	-	-	-	0.04	-	-
11	Benzaldehyde	Other (benzene derivative)	968	-	0.44	0.03	0.21	0.11	0.12	0.09	0.15
22	Phenylacetaldehyde	Other (benzene derivative)	1050	0.08	0.38	0.15	0.24	0.05	0.12	0.12	0.14
25	Acetophenone	Other (benzene derivative)	1072	-	0.13	-	-	-	0.05	-	-
39	2,4-Dimethylbenzaldehyde	Other (benzene derivative)	1179	-	0.17	-	-	-	-	-	-
40	*p*-Methylacetophenone	Other (benzene derivative)	1190	-	-	-	-	-	-	-	0.13
41	3,4-Dimethylphenol	Other (benzene derivative)	1197	-	-	0.05	-	-	-	0.09	-
45	Indole	Other (benzene derivative)	1296	-	0.32	0.05	0.14	-	0.07	0.1	-
72	Benzophenone	Other (benzene derivative)	1630	-	-	-	-	-	-	0.17	0.18
90	Diisobutyl phthalate	Other (benzene derivative)	1872	-	0.47	-	0.41	0.07	0.92	0.32	0.95
96	Dibutyl phtalate	Other (benzene derivative)	1967	-	0.25	-	0.66	-	1.5	-	0.23
33	4-Ketoisophorone	Other (C_13_-norisoprenoid)	1149	-	0.24	0.03	0.09	-	-	-	-
52	α-Ionone	Other (C_13_-norisoprenoid)	1433	0.11	-	-	-	-	0.04	-	-
56	β-Ionone	Other (C_13_-norisoprenoid)	1489	0.89	3.1	0.32	0.83	0.25	0.57	0.57	0.49
65	Dodecanoic acid	Other (carboxylic acid)	1573	-	-	-	-	-	-	0.16	-
97	Palmitic acid	Other (carboxylic acid)	1973	-	2.35	-	1.21	-	-	-	0.82
108	Heptadecanoic acid	Other (carboxylic acid)	2103	-	1.00	-	0.34	0.1	0.3	0.34	0.29
109	(*Z*,*Z*)-Octadeca-9,12-dienoic acid	Other (carboxylicacid)	2110	-	0.3	0.23	0.31	0.21	0.2	0.62	0.51
112	Oleic acid	Other (carboxylic acid)	2183	-	-	-	-	-	0.27	0.85	0.4
32	Dictyopterene A (1-[(1*E*) -hexenyl]-2-vinylcyclopropane])	Other (dictyopterene)	1122	0.07	-	0.08	-	-	-	-	-
34	Dictyopterene D’ (6-[(1*Z*)-butenyl]-1,4-cycloheptadiene])	Other (dictyopterene)	1157	0.11	-	0.06	-	-	-	-	-
38	Dictyopterene C’ ([6-butyl-1, 4-cycloheptadiene])	Other (dictyopterene)	1174	-	-	0.05	-	-	-	-	-
100	Methyl octadecyl ether	Other (ether)	2032	2.21	-	-	-	1.28	1.34	1.55	1.42
17	2-Pentylfuran	Other (furan)	994	-	0.6	0.06	0.2	-	0.08	0.09	0.15
102	Methyl (all *Z*) 5,8,11,14-eicosatetraenoate	Other (long fatty acid ester)	2049	0.69	0.24	0.83	0.29	0.66	-	1.29	0.56
103	Methyl (all *Z*) 5,8,11,14,17-eicosapentaenoate	Other (long fatty acid ester)	2056	2.95	0.24	1.41	0.32	0.92	-	0.98	0.37
107	(*E*)-9-Octadecenoic acid methyl ester	Other (long fatty acid ester)	2090	-	-	-	0.18	-	-	-	-
115	Ethyl icosanoate	Other (long fatty acid ester)	2398	6.14	12.85	30.84	29.59	34.51	30.73	10.2	25.37
42	β-Cyclocitral	Other (monoterpene aldehyde)	1224	-	0.35	0.04	0.1	-	-	-	-
44	β-Cyclohomocitral	Other (monoterpene aldehyde)	1263	-	0.16	-	-	-	-	-	-
53	(Z)-Geranylacetone	Other (monoterpene ketone)	1458	-	0.22	-	-	-	-	-	-
**Total identified=**				**81.62**	**87.59**	**81.88**	**85.85**	**81.21**	**87.84**	**83.71**	**89.11**

No—order of the compounds elution on HP-5MS column; RI—retention index; SAC—Saturated aliphatic compound; UnSAC—Unsaturated aliphatic compound.

**Table 4 marinedrugs-21-00415-t004:** Major non-volatile compounds in *Cladostephus spongiosus* ethanol extract identified using high performance liquid chromatography–high-resolution mass spectrometry with electrospray ionisation (UHPLC-ESI–HRMS).

No.	Name	Monoisotopic Mass (Da)	[M + H]^+^	Molecular Formula	t_R_ (min)	Mass Difference (ppm)	Area (Counts)
**FATTY ACID DERIVATIVES**							
3	1-(9*Z*,12*Z*,15*Z*-Octadecatrienoy)l-3-O-β-D-galactosyl)-sn-glycerol	514.31418	515.32146	C_27_H_46_O_9_	10.94	1.7	1.73 × 10^6^
4	1-(9*Z*,12*Z*,15*Z*-Octadecatrienoy)l-3-O-(6′-O-α-D-galactosyl-β-D-galactosyl)-sn-glycerol	676.36701	677.37428	C_33_H_56_O_14_	10.94	3.0	1.00 × 10^6^
5	1-(9*Z*,12*Z*-Octadecadienoy)l-3-O-(6′-O-α-D-galactosyl-β-D-galactosyl)-sn-glycerol	678.38266	679.38993	C_33_H_58_O_14_	11.42	1.2	1.39 × 10^6^
6	1-(9*Z*-Octadecenoy)l-3-O-(6′-O-α-D-galactosyl-β-D-galactosyl)-sn-glycerol	680.39831	681.40558	C_33_H_60_O_14_	11.99	2.6	9.37 × 10^5^
7	Myristamide (tetradecanamide)	227.22491	228.23219	C_14_H_29_NO	12.47	4.3	2.76 × 10^7^
8	Palmitoleamide (hexadec-9-enamide)	253.24056	254.24784	C_16_H_31_NO	12.96	3.1	9.19 × 10^7^
9	Linoleamide (octadeca-9,12-dienamide)	279.25621	280.26349	C_18_H_33_NO	13.42	3.6	8.71 × 10^7^
10	Palmitamide (hexadecanamide)	255.25621	256.26349	C_16_H_33_NO	13.68	3.5	1.28 × 10^8^
11	2-(5*Z*,8*Z*,11*Z*,14*Z*-Eicosatetraenoyl)-sn-glycerol	378.27701	379.28429	C_23_H_38_O_4_	13.77	0.4	1.54 × 10^6^
12	Glyceryl palmitate (2,3-dihydroxypropyl hexadecanoate)	330.27701	331.28429	C_19_H_38_O_4_	13.99	3.5	1.08 × 10^7^
13	Oleamide (octadec-9-enamide)	281.27186	282.27914	C_18_H_35_NO	14.10	4.5	1.27 × 10^7^
14	Glyceryl monooleate (2,3-dihydroxypropyl-octadec-9-enoate)	356.29266	357.29994	C_21_H_40_O_4_	14.37	1.6	5.00 × 10^6^
17	Stearamide (octadecanamide)	283.28751	284.29479	C_18_H_38_NO	14.79	4.4	4.37 × 10^7^
18	Glyceryl monostearate (2,3-dihydroxypropyl octadecanoate)	358.30831	359.31559	C_21_H_42_O_4_	15.06	0.8	9.93 × 10^6^
19	Gondamide (icos-11-enamide)	309.30316	310.31044	C_20_H_39_NO	15.08	3.0	3.52 × 10^7^
20	Arachidonic acid (icosa-5,8,11,14-tetraenoic acid)	304.24023	305.24751	C_20_H_32_O_2_	15.34	3.0	1.60 × 10^7^
24	Erucamide (docos-13-enamide)	337.33446	338.34174	C_22_H_43_NO	15.98	3.7	3.01 × 10^7^
**PIGMENTS AND DERIVATIVES**							
1	Loliolide	196.10994	197.11722	C_11_H_16_O_3_	6.14	5.2	3.30 × 10^6^
16	Fucoxanthin	658.42334	659.43062	C_42_H_58_O_6_	14.77	2.3	2.98 × 10^7^
15	Halocynthiaxanthin acetate	640.41277	641.42005	C_42_H_56_O_5_	14.77	2.3	1.86 × 10^6^
21	Pheophorbide *a*	592.26857	593.27585	C_35_H_36_N_4_O_5_	15.37	2.8	7.44 × 10^7^
22	3-[(21*R*)-21-(Methoxycarbonyl)-4,8,13,18-tetramethyl-20-oxo-9,14-divinyl-3,4-didehydro-3-24,25-dihydrophorbinyl]propanoic acid	588.23727	589.24455	C_35_H_32_N_4_O_5_	15.60	1.7	9.32 × 10^6^
23	(2*E*)-3-[21-(Methoxycarbonyl)-4,8,13,18-tetramethyl-20-oxo-9,14-divinyl-3,4-didehydro-3-24,25-dihydrophorbinyl]acrylic acid	586.22162	587.2289	C_35_H_30_N_4_O_5_	15.67	0.9	3.11 × 10^6^
25	4-{[(2*R*,3*S*,4*S*,5*R*,6*R*)-6-{[(2*S*,3*S*,4*S*,5*R*)-5-({[3-Carboxy-3-(dodecylamino)propanoyl]oxy}methyl)-3,4-dihydroxy-2-(hydroxymethyl)tetrahydro-2-furanyl]oxy}-3,4,5-trihydroxytetrahydro-2H-pyran-2-yl]methoxy}-2 -(dodecylamino)-4-oxobutanoic acid	908.54570	909.55298	C_44_H_80_N_2_O_17_	16.06	4.4	2.12 × 10^4^
27	Pheophytin *b*	884.54519	885.55246	C_55_H_72_N_4_O_6_	18.97	11.0	3.11 × 10^5^
28	Loroxanthin decenoate	736.54309	737.55034	C_50_H_72_O_4_	19.06	3.3	6.68 × 10^4^
**STEROID DERIVATIVES**							
2	β-Sitosterol 3-O-acetate	456.39673	457.40401	C_31_H_52_O_2_	9.92	4.5	2.13 × 10^6^
26	Fucosterol epoxide	428.36543	429.37271	C_29_H_48_O_2_	17.41	1.0	1.98 × 10^6^

No—order of the compounds elution on Acquity UPLC BEH Phenyl-Hexyl analytical column.

## Data Availability

Data are contained within the article.

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
