# Peer review of "Seasonal Monitoring of Volatiles and Antioxidant Activity of Brown Alga Cladostephus spongiosus"

_marinedrugs, 2023, doi:10.3390/md21070415_

Round 1
Reviewer 1 Report
Some suggestions:
Abstract: 1) How often the samples were collected? Monthly sampling? please state in the abstract, 2) what does the season refer to?
Introduction: 1) How to determine the growth period of the algae? 2) Please add how the season and extraction methods affect the antioxidant activity
Results: 1) Most references were missing, please revise. 2) Please make the graph more contrast, between Fr/Dr or month. 3) What compounds are the most abundant and the specific compounds for each season. 4) How about the yield of the extracts? 5) Which compounds are responsible for the antioxidant activity. 6) Please write scientific name correctly
Conclusion: Please write concisely.
Reviewer 2 Report
In the present work, a wide series of volatile organic compounds (VOC) of different types (alcohols, aldehydes, terpenes...) in extracts of brown algae, C. spongious, have been identified through mass spectrometry (MS) analysis. To reach the identification of the compounds by comparison in mass libraries, two types of extractions have been carried out, in different samples of algae. On the other hand, the antioxidant activity of the extracts has been evaluated in three types of trials.
The summary of the work indicates that VOCs have been isolated, which is not correct, because no pure compound or a mixture has been obtained. What the work describes is the relative identification by means of MS, by bibliographic comparison:
Line 10: It is proposed to the authors to change the word isolation.
In general, a revision of the abstract is proposed to the authors, which should not be a list of objectives (line 10-16), in addition, it is repeated at the end of the introduction (line 69-76).
The references are badly cited with numerous errors, a deep review is suggested both in the text and at the end of the references.
Regardless of the quality of English, from the grammatical point of view, it is difficult to understand and understand the research carried out, because the results are not well organized, since there is a long list of identified compounds of different nature, to facilitate the For the reader's understanding, it is suggested that the compounds be grouped by type (Tables 1, 2, 3), as represented in the figures (Figure 1, 2, 3). On the other hand, it is not clear what criteria the authors have chosen to enumerate the compounds in the tables, since they do not indicate it. Nor does the meaning of the initials RI appear in the work (Tables 1, 2, 3).
In Figure 1, the authors do not indicate that they measure on the y axis.
The value of the trolox control is lacking in antioxidant activity assays.
In Table 4, the area data indicated by the authors is not useful, since it is an arbitrary value that cannot be compared with other studies. It is suggested to give a relative percentage or a comparable value e.g. ng/g dry weight, microg/g dry weight…
Finally, throughout the text there are numerous names of algae that are not in italics and it is suggested that they be corrected.
Round 2
Reviewer 1 Report
The manuscript has been improved. However, the conclusion can be more direct and concise.
Author Response
The conclusion was rewritten concisely.
Reviewer 2 Report
Thank you for considering changing the word “isolated”. I think it would have been more accurate to put "identified", not "obtained". Line 529 The reference is badly cited, the title is missing. Finally, regarding the presentation of the results, it may seem subjective, but the main thing in the transfer of scientific knowledge is to obtain the most information in a clear and simple way, in this case it is laborious. I believe that the presentation of the data in table 4 is more accurate, grouping by types of compounds, compared to the results of tables 1,2,3 by retention time, due to the high number of compounds.
Author Response
The word "obtained" has been changed to "identified".
All cited references are checked once again.
The components in Tables 1-3 are now grouped by structural type.